# Few-shot Semantic Segmentation via Perceptual Attention and Spatial Control

## ABSTRACT

Few-shot semantic segmentation (FSS) aims to locate pixels of unseen classes with clues from a few labeled samples. Recently, thanks to profound prior knowledge, diffusion models have been expanded to achieve FSS tasks. However, due to probabilistic noising and denoising processes, it is difficult for them to maintain spatial relationships between inputs and outputs, leading to inaccurate segmentation masks. To address this issue, we propose a Diffusion-based Segmentation network (DiffSeg), which decouples probabilistic denoising and segmentation processes. Specifically, DiffSeg leverages attention maps extracted from a pretrained diffusion model as support-query interaction information to guide segmentation, which mitigates the impact of probabilistic processes while benefiting from rich prior knowledge of diffusion models. In the segmentation stage, we present a Perceptual Attention Module (PAM), where two cross-attention mechanisms capture semantic information of support-query interaction and spatial information produced by the pretrained diffusion model. Furthermore, a self-attention mechanism within PAM ensures a balanced dependence for segmentation, thus preventing inconsistencies between the aforementioned semantic and spatial information. Additionally, considering the uncertainty inherent in the generation process of diffusion models, we equip DiffSeg with a Spatial Control Module (SCM), which models spatial structural information of query images to control boundaries of attention maps, thus aligning the spatial location between knowledge representation and query images. Experiments on PASCAL-$5^i$ and COCO datasets show that DiffSeg achieves new state-of-the-art performance with remarkable advantages.

## CCS CONCEPTS

• **Computing methodologies** → **Image segmentation**; **Machine learning**.

## KEYWORDS

Few-shot segmentation, Diffusion model, Perceptual attention, Spatial control

## 1 INTRODUCTION

Semantic segmentation has achieved tremendous success due to the advancement of deep learning methods. However, deep learning

*ACM MM, 2024, Melbourne, Australia*
© 2024 Copyright held by the owner/author(s). Publication rights licensed to ACM.
ACM ISBN 978-x-xxxx-xxxx-x/YY/MM
https://doi.org/10.1145/nnnnnnn.nnnnnnn

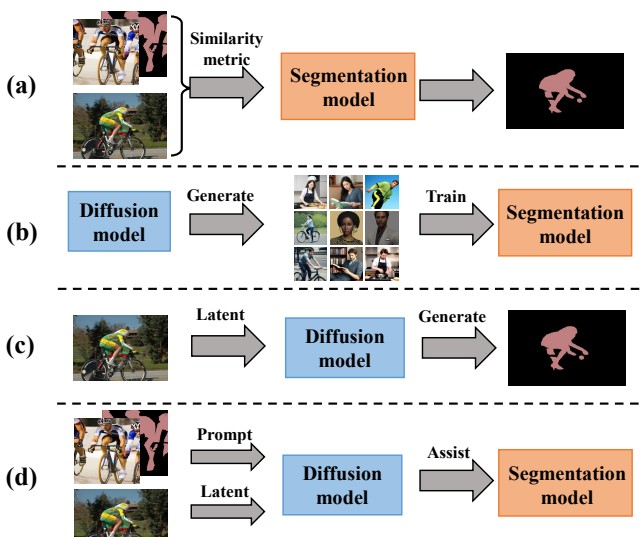

**Figure 1: Comparison among different models. (a) Metric-based methods find co-occurrence features between support images and query images to locate targets. (b) Generation-based methods synthesize adequate samples for training segmentation models. (c) Regarding segmentation masks as generated targets, Diffusion segmenters directly perform segmentation task during the denoising process. (d) Our proposed DiffSeg decouples probabilistic denoising and segmentation processes, which could reduce the impact of probabilistic processes while benefiting from profound prior knowledge of pretrained diffusion models.**

algorithms require large amounts of dataset to learn knowledge, where labeling is extremely time-consuming and annoying. Essentially, humans could understand a new category with a few samples, which inspires researchers to transfer knowledge from known to unknown. By properly encoding humans' knowledge, few-shot learning successfully relieves the high demand of collecting large-scale dataset, thus expanding its usage into variant scenarios [1, 6, 14].

In this paper, we focus on the task of Few-shot Semantic Segmentation (FSS), which aims to segment new categories with a few samples. Recently, metric-based methods [5, 17] have significantly contributed to the FSS task, which locate targets by finding co-occurrence features between support images and query images, as shown in Fig.1 (a). However, these methods face challenges in fully leveraging intrinsic features of categories with limited samples, resulting in compromises in target integrity and boundary accuracy. Thanks to profound prior knowledge, diffusion models [26] can thoroughly express intrinsic features of new categories.

Nevertheless, extended to FSS, those methods [27] are merely used to generate samples for training segmentation models, as shown in Fig.1 (b). It essentially reverts FSS tasks back to conventional segmentation tasks, which fails to meet the requirements of low time consumption and computational cost for training. To avoid this issue, researchers propose diffusion segmenters [12? ], which directly perform the segmentation task during denoising processes, as shown in Fig.1 (c). Due to probabilistic noising and denoising processes, those methods struggle to maintain spatial relationships between inputs and outputs, thus generating results with inaccurate boundaries.

Facing the aforementioned challenges, we propose a novel FSS paradigm named DiffSeg, as shown in Fig.1 (d). DiffSeg utilizes prior knowledge of a pretrained diffusion model to assist in segmentation, thus mining intrinsic features of categories with limited samples and avoiding generating probabilistic results. Specifically, we feed support images as prompts and query images as latent images into a diffusion model, where we extract multiscale attention maps, i.e., self-attention maps and cross-attention maps. The former effectively capture the similarity relationship among pixels, which helps understand semantic contents of query images. The latter contain response information of support-query interaction, which could help the model locate target areas. In fact, the attention maps are able to locate the general position of target areas through simply processing, which provide a strong guidance for segmentation.

In the segmentation phase, we propose a Perceptual Attention Module (PAM) to highlight significant regions within query images based on support images and attention maps of a pretrained diffusion model, respectively. Firstly, we employ two cross-attention mechanisms to handle data of different branches. The cross-attention between query-attention pairs could capture spatial information produced by pretrained diffusion models, thus introducing prior knowledge of pretrained diffusion models into segmentation phase. Simultaneously, the cross-attention between query-support images could capture semantic interaction information, which could find co-occurrence objects between support images and query images. Secondly, considering confliction between the spatial information and the semantic information, we utilize a self-attention to establish a balanced dependence for segmentation, which improves the robustness of DiffSeg. Noted that those computations are conducted at the latent image level. A latent image is a compression transformation that attenuates high frequency information of a natural image, which not only reduces computational burdens, but also keeps visual features of the original natural image.

Considering the generation uncertainty of diffusion models, we propose a Spatial Control Module (SCM) for DiffSeg to align semantic boundaries between extracted attention maps and query images. Specifically, we integrate edge maps of query images with the hidden states of down-sample processes in diffusion models. Subsequently, this combined information is transformed into spatial structural information and inserted into corresponding scale levels of up-sample processes. On the one hand, the spatial structural information serves as conditional input to control spatial structural information of attention maps during up-sample processes. On the other hand, direct connections between corresponding down-sample and up-sample processes help mitigate changes in spatial structures of attention maps. Differing from ControlNet [44], which employs latent images for control, SCM utilizes conditional information to govern attention maps.

The main contributions of this paper are as follows:

- We introduce a novel FSS paradigm named DiffSeg, which effectively decouples probabilistic processes of diffusion models with segmentation phases, thus avoiding uncertain results while utilizing profound prior knowledge.
- We propose a Perceptual Attention Module for segmentation, where two cross-attention capture semantic information of support-query interaction and spatial information produced by the pretrained diffusion model, respectively, Additionally, a self-attention mechanism is employed to find a balance in dependency for segmentation tasks.
- Considering the probabilistic generation of diffusion models, we propose a Spatial Control Module to align semantic boundaries between extracted attention maps and query images, which keeps spatial structures of query images during diffusion processes.
- Experimental results on the PASCAL-$5^i$ dataset show the proposed method achieves the mIOU score of 69.3% for 1-shot segmentation and 72.1% for 5-shot segmentation, setting new state-of-the-art performance with remarkable advantages.

## 2 RELATED WORK

### 2.1 Semantic Segmentation

Semantic segmentation aims to classify each pixel of an image into a set of preset categories. According to different network structures, current methods can be roughly divided into four categories, i.e., CNN-based, RNN-based, GNN-based and transformer-based methods. CNN-based methods [13, 18, 43] utilize convolution operations to extract semantic information from feature maps for pixel-level label prediction. Considering dependence of context information, RNN-based methods [9, 22, 32] use recurrent layers to capture local and global spatial structure information of images. Using topological structure of graphs, GNN-based methods [19, 20, 30] transform task of image segmentation into the classification task of graph nodes. Recently, transformer-based methods [37, 39, 46] have received more popularity. They regard patches of images as a sequence, then utilize an encoder-decoder structure with attention mechanisms to achieve segmentation.

Among them, transformer-based methods are most relevant to our work. To comprehensively understand image contents, IncepFormer [4] introduces an efficient inception transformer that integrates global context, fine localization information and multiscale features to segment images. Considering the background incompleteness issue, Liu et al. [16] propose WegFormer, where the depth-taylor decomposition principle and soft erasure module are incorporated to generate more complete pseudo-labels. Using a pretrained transformer in a data collection loop, Kirillov et al. [11] propose a Segment Anything (SA) project: a new task, model, and dataset for image segmentation, which shows the advantages of transformer for image segmentation tasks.

### 2.2 Few-shot Semantic Segmentation

Few-shot semantic segmentation extends the ability of segmentation to novel category with a few labeled samples. Many researchers

regard few-shot segmentation as a guided segmentation task, thus following a two-branch framework [24, 41]. For instance, Shaban et al. [28] apply few-shot learning on semantic segmentation using a two-branch framework, where the support branch generates parameters in the last layer of the query branch for segmentation. Following their idea, Michaelis et al. [21] combine embeddings with a U-net to find unknown objects in a complex scene guided by only one sample. To better utilize information of the support set, Wang et al. [34] learn prototype representations based on a few support images in an embedding space, which could match pixels to the learned prototypes, thus performing segmentation in few-shot settings.

Recently, Gu et al. [5] propose DRCNet to achieve sufficient support-query interaction for accurate FSS, where a dynamic context module is presented to capture spatial details in query images by building dynamic convolutions in local views. For better feature fusion, MFNet [45] utilizes an attention mechanism to achieve support feature modulation and multi-scale combination. Facing heavy computational operations of high-dimensional vectors, QCLNet [47] explores latent interaction between images through the utilization of operations grounded in well-established quaternion algebra..

However, those methods are challenging in digging internal relationship between query and support images, due to limited available semantic information with a few samples. Unlike former methods, DiffSeg introduces a pretrained diffusion model to provide sufficient prior knowledge for guiding segmentation, thus reducing the dependence on training data scale.

## 2.3 Diffusion Models for Segmentation

Diffusion models have recently gained significant attention from the research community due to their ability to generate high-fidelity contents, which are gradually extended to FSS tasks [2, 33]. For instance, to effectively train segmentation models using generated images, Roy et al. [27] propose diffusion-based DiffAlign, which could align the synthetic images to the real images and minimize the domain gap.

In contrast to those methods that synthesize samples, LEDM [2] directly use the generative model to perform segmentation, which extracts the intermediate activations from the reverse diffusion process as excellent pixel-level representations for the segmentation problem. In order to enhance the step-wise regional attention in diffusion probabilistic model for the image segmentation, MedSegDiff [36] proposes dynamic conditional encoding, which establishes the state-adaptive conditions for each sampling step.

Different from previous methods, our method decouples probabilistic processes and segmentation phase, which could reduce the impact of probabilistic processes while benefiting from rich prior knowledge of pretrained diffusion models.

## 3 TASK DESCRIPTION

Given only one or a few images with pixel-level annotations, few-shot segmentation aims to accurately locate foreground pixels in test images. Specifically, we divide the dataset into training set $C_{train}$ and test set $C_{test}$ according to category of images, where $C_{train}$ and $C_{test}$ don't contain same-category images, which can be represented as $C_{train} \cap C_{test} = \varnothing$.

For each $k$-shot segmentation task, we firstly define a target class $c$, and then sample $k + 1$ images with the class label $c$ from $C_{test}$. Defining the first $k$ labeled images as support set $S$ and the last image as $x_q$, few-shot segmentation model $M_\theta$ aims to compute the segmentation mask of the query image $\hat{y}_q$ with:

$$\hat{y}_q = M_\theta(S, x_q) \tag{1}$$

where $\theta$ are parameters of the model. Assuming that $y_q$ is the ground-truth label for the query image $x_q$, task goal of few-shot segmentation is to minimize the loss $L(\hat{y}_q, y_q)$ by updating $\theta$, where such process could be represented as:

$$\theta = \arg\min_\theta L(\hat{y}_q, y_q) = \arg\min_\theta L(M_\theta(S, x_q), y_q) \tag{2}$$

Since an image may contain objects of different categories, the ground-truth query mask could vary with different assigned labels.

## 4 METHOD

### 4.1 Overview

The framework of DiffSeg is shown in Fig.2, where a Perceptual Attention Module (PAM) performs perceptual attention mechanisms among support latent images, query latent images and attention maps generated by a knowledgeable diffusion model, which helps locate target areas with compact boundaries. Besides, a Spatial Control Module (SCM) inserts conditional spatial information into the denoising process of diffusion UNet, which aligns edges between query images and attention maps, thus keeping spatial structures of query images during probabilistic diffusion processes.

Specifically, we first multiply a support image $I_s$ and a binary support mask $M_s$ to remove background. Then, the output is processed by a Variational Auto-Encoder (VAE) encoder and a Clip [23] to generate the support latent image $L_s$ and embedding $E_m$, respectively. The process can be expressed as:

$$L_s = Encode(I_s \otimes M_s) \tag{3}$$

$$E_m = Clip(I_s \otimes M_s) \tag{4}$$

where $\otimes$ is element-wise multiplication. The embedding $E_m$ generated by Clip is the conditional information of diffusion models, which is interacted with query latent images during denoising processes. Then encoder module transforms the query image $I_q$ to the query latent image $L_q$, which is input into a diffusion model prompted by $E_m$ to obtain attention maps $A_m$. Noted that $I_s$ is processed by HDE algorithm [38] to generate an edge map $I_l$, which is used by SCM to control attention maps of diffusion model, thus aligning spatial structures between attention maps and query images. The process can be expressed as:

$$L_q = Encode(I_q) \tag{5}$$

$$A_m = Diff(L_q|E_m, SCM(I_l)) \tag{6}$$

where $Diff(\cdot)$ and $SCM(\cdot)$ represent the process of diffusion model and spatial control module, respectively. Then, the attention maps $A_m$ is processed to obtain an attention score map $M_a$:

$$M_a = Process(A_m) \tag{7}$$

where $Process(\cdot)$ presents the operation that transforms $A_m$ to $M_a$, which details are shown in section 4.2.

Afterwards, PAM performs perceptual attention mechanisms among attention maps $M_a$, the support latent image $L_s$ and the

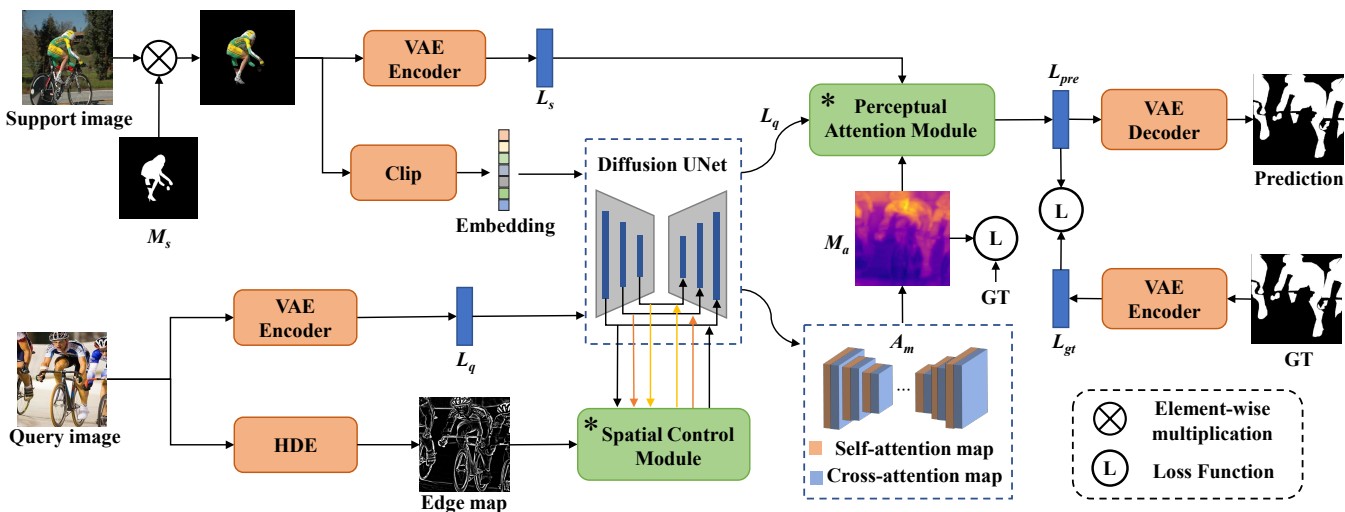

**Figure 2: The pipeline of DiffSeg, where * means updating parameters in training phase. Firstly, a pretrained diffusion model extracts support-query interaction information $M_a$, which provides a strong guidance for segmentation. Then, PAM emphasizes key areas of query image based on support images and $M_a$, thus achieving dense prediction with compact boundaries. In this manner, DiffSeg decouples probabilistic de-noising and segmentation processes, thus reducing the impact of probabilistic processes while benefiting from profound prior knowledge of diffusion models.**

query latent image $L_q$ to locate target areas with compact boundaries, which outputs the predicted latent image $L_{pre}$.

$$L_{pre} = PAM(M_a, L_s, L_q) \qquad (8)$$

Finally, a VAE decoder transforms $L_{pre}$ to prediction image $I_{pre}$, which can be formulated as:

$$I_{pre} = Decode(L_{pre}) \qquad (9)$$

In training phase, we only update parameters of SCM and PAM. Therefore, we supervise the attention score map $M_a$ and the predicted latent image $L_{pre}$, where both two supervision use the MSE loss function. For computing loss of $L_{pre}$, we use a VAE [10] encoder to obtain the latent image of ground truth. The whole loss $\mathcal{L}$ can be computed as:

$$\mathcal{L} = \alpha L(M_a, I_{gt}) + \beta L(L_{pre}, Encode(I_{gt})) \qquad (10)$$

where $L(\cdot)$ presents MSE loss function and $I_{gt}$ refers to ground truth. $\alpha$ and $\beta$ are weights of each loss. Based on experimental study shown in supplementary, we set them to 2 and 1, respectively.

For $k$-shot segmentation, both $k$ embeddings and $k$ support latent images are fused in an average manner. Thanks to the ability of Clip to align image with text and rich prior knowledge of pretrained diffusion models, there is little gap in segmentation performance between different fusion methods. Experiments in supplementary can prove it. Therefore, we select simple but effective average fusion as our $k$-shot solution.

### 4.2 Diffusion UNet

To incorporate profound prior knowledge into the FSS task, we equip DiffSeg with a pretrained diffusion model named Kandinsky [25], which helps understand semantic contents of query images

and achieve support-query interaction. Specifically, we input support images as prompt and query images as latent into Kandinsky, and then extract attention maps from the model, i.e., cross-attention maps and self-attention maps. The former contain the response information of query images to support images, and the latter helps fully extract semantic information from query images.

Kandinsky diffusion contains attention maps of 3 different scales (i.e., 8×8, 16×16, 32×32), which are interpolated to the same size for fusion. By averaging self-attention maps and cross-attention maps, respectively, we can obtain a fused self-attention map $M_{self} \in \mathbb{R}^{HW \times HW}$ and a fused cross-attention map $M_{cross} \in \mathbb{R}^{H \times W}$.

In fact, cross-attention maps serve as a vital component in the conditional generation process, providing valuable insights into the conditional probability distribution. However, the produced cross-attention score maps often lack clear object boundaries and may exhibit internal holes. Fortunately, self-attention maps are able to establish the correlations between different pixels, which provides the ability to perform region completion, thus compensating for the incomplete activation regions in cross-attention. Therefore, the final attention score map $A_m$ can be obtained by multiplying the cross-attention maps with pixel affinity weights obtained from self-attention maps:

$$M_a = norm(M_{self} \cdot vec(M_{cross})) \qquad (11)$$

where $norm(\cdot)$ is min-max normalization to ensure the segmentation score maps are appropriately scaled, $vec(M_{cross}) \in \mathbb{R}^{HW \times 1}$, and $vec(\cdot)$ is a vectorization operation of a matrix.

Briefly, the attention score map $A_m$ is simply produced during the denoising inference process of the pretrained diffusion model, which locates target areas with prior knowledge of the diffusion model, thus providing useful guidance for segmentation phases.

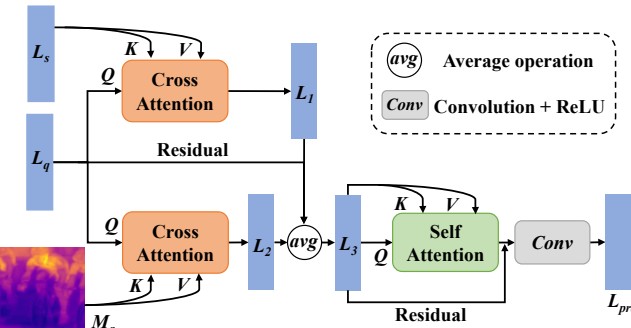

Figure 3: The design of perceptual attention module. Two cross-attention operations provide segmentation task for semantic information of support-query interaction and spatial information produced by prior information of pretrained diffusion models, respectively. Considering confliction between them, we utilize a self-attention to find a balance of dependence for segmentation.

### 4.3 Perceptual Attention Module

Different from most previous methods which performs dense comparisons within feature maps of images, PAM performs perceptual attention mechanisms in latent image level. A latent image is a compression transformation that removes high frequency information of a natural image, which not only reduces computational burdens in comparison, but also keeps visual features of original images. Moreover, PAM considers extracted attention maps to perceptual attention, thus introducing prior knowledge of diffusion models into segmentation phase.

The design of PAM is shown in Fig.3. Firstly, we perform cross-attention for a query latent image $L_q$ with a support latent image $L_s$ and an attention score map $M_a$, respectively. In this process, we set $L_q$ as Query ($Q$) and set $L_s$ or $M_a$ as Key ($K$) and Value ($V$). The cross-attention can be computed as:

$$L_1 = Softmax(\frac{Liner(L_q) \cdot Liner(L_s)}{\sqrt{d}}) \cdot Liner(L_s) \qquad (12)$$

where $Liner(\cdot)$ function refers to linear transformation and $d$ presents the dimension of linear features. In the same manner, cross-attention is performed between $L_q$ and $M_a$ to obtain $L_2$.

Essentially, $L_1$ contains semantic information of support-query interaction, while $L_2$ contains spatial information produced by the pretrained diffusion model. In most cases, they have the same interested areas, which could provide positive guidance for segmentation. Unfortunately, their interested areas are completely inconsistent on occasion.

Considering conflict between the prior information and the semantic interaction information, we utilize a self-attention to find a balance of dependence for segmentation. Specifically, we fuse $L_q$, $L_1$ and $L_2$ in an average manner to achieve $L_3$, which is set to $K$, $Q$ and $V$ in the self-attention operation. Then, the output of self-attention is incorporated to $L_3$ in a residual form. Finally, we use a convolution and ReLU block to achieve the latent image $L_{pre}$ of predicted mask.

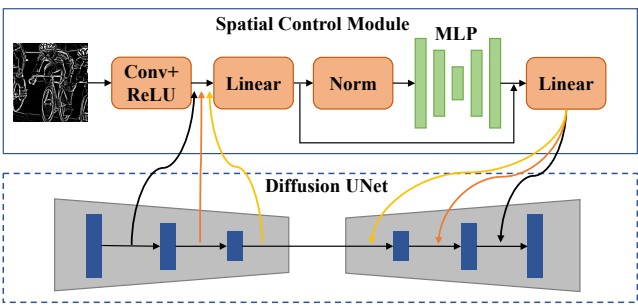

Figure 4: The design of spatial control module, where arrows of different colors represent different paths. SCM preserves the capabilities of the pretrained diffusion model by freezing its parameters, and inserts spatial conditional information into the model to control its semantic boundaries.

### 4.4 Spatial Control Module

Due to probabilistic noising and denoising processes, diffusion models are unable to maintain spatial relationships between inputs and outputs, leading to inaccurate segmentation masks. To solve this problem, we present an SCM for DiffSeg to align semantic boundaries between extracted attention maps and query images, which is shown in Fig.4.

SCM treats the pretrained diffusion model as a strong backbone for learning conditional controls. Firstly, an edge map is processed by convolution and ReLU blocks to obtain spatial information of a query image. Then it merges with features resized to $64 \times 64$ of diffusion UNet and goes through a linear layer. The output of the linear layer not only goes through a normalized layer and a Multi-Layer Perception (MLP), but is also added to output of MLP in a residual form to avoid excessive transforms. Finally, after the process of a linear layer, we insert spatial controls into diffusion UNet in a residual form, which helps align spatial structures between attention maps and query images. On the one hand, the spatial information is regarded as conditional information to control spatial structures of attention maps in up-sample processes. On the other hand, direct connections between corresponding down-sample and up-sample processes could reduce changes in spatial structures. Noted that SCM extracts features from each scale level of UNet and inserts spatial controls into a corresponding level, as shown in Fig.4, where arrows of different color represent different paths.

Differing from ControlNet [44], which employs latent images for control, SCM utilizes conditional information to govern attention maps Additionally, while ControlNet operates at each encoder level of a diffusion U-net with cascaded multiscale encoder blocks, SCM handles state information from various levels by parameter sharing, thus achieving control with reduced computational overhead.

## 5 EXPERIMENTS

### 5.1 Dataset and Evaluation Metric

*PASCAL-5$^i$* [28] is a public dataset of few-shot semantic segmentation, which is made up of PASCAL VOC 2012 [3] with extra SDS [7] annotations. The dataset includes 20 categories, which are divided into 4 splits, and each split contains 5 categories.

Table 1: Comparison results of 1-shot and 5-shot Segmentation on PASCAL-$5^i$. Best in Bold and Second in Underline.

| Method | mIOU (1-shot) | | | | | FB-IOU (1-shot) | mIOU (5-shot) | | | | | FB-IOU (5-shot) |
|---|---|---|---|---|---|---|---|---|---|---|---|---|
| | split-1 | split-2 | split-3 | split-4 | mean | | split-1 | split-2 | split-3 | split-4 | mean | |
| PFENet [31] | 61.7 | 69.5 | 55.4 | 56.3 | 60.8 | 73.3 | 63.1 | 70.7 | 55.8 | 57.9 | 61.9 | 73.9 |
| DCAMA [29] | 67.5 | 72.3 | 59.6 | 59.0 | 64.6 | 75.7 | 70.5 | 73.9 | 63.7 | 65.8 | 68.5 | 79.5 |
| CyCTR [42] | 65.7 | 71.0 | 59.5 | 59.7 | 64.0 | 74.3 | 69.3 | 73.5 | 63.8 | 63.5 | 67.5 | 75.9 |
| DRCNet [5] | 70.3 | 74.7 | 67.9 | 62.0 | 68.7 | 75.8 | 72.3 | 76.5 | 70.6 | 68.2 | 71.9 | 77.4 |
| AAFormer [35] | 69.1 | 73.3 | 59.1 | 59.2 | 65.2 | 73.8 | 72.5 | 74.7 | 62.0 | 61.3 | 67.6 | 76.2 |
| VAT [8] | 67.6 | 72.0 | 62.3 | 60.1 | 65.5 | 77.8 | 72.4 | 73.6 | 68.6 | 65.7 | 70.1 | 80.9 |
| DPCN [17] | 65.7 | 71.6 | **69.1** | 60.6 | 66.7 | 78.0 | 70.0 | 73.2 | **70.9** | 65.5 | 69.9 | 80.7 |
| MIANet [40] | 68.5 | 75.7 | 67.4 | 63.1 | 68.6 | 76.3 | 70.2 | 77.3 | 70.0 | 68.8 | 71.6 | 76.8 |
| DiffSeg (ours) | **70.5** | **75.8** | 67.9 | **63.2** | 69.3 | 78.4 | 72.8 | 77.6 | 68.7 | 69.3 | 72.1 | **81.2** |

Table 2: Comparison results of 1-shot and 5-shot Segmentation on COCO dataset.

| Method | mIOU (1-shot) | | | | | FB-IOU (1-shot) | mIOU (5-shot) | | | | | FB-IOU (5-shot) |
|---|---|---|---|---|---|---|---|---|---|---|---|---|
| | split-1 | split-2 | split-3 | split-4 | mean | | split-1 | split-2 | split-3 | split-4 | mean | |
| PFENet [31] | 36.8 | 41.8 | 38.7 | 36.7 | 38.5 | 63.0 | 40.4 | 46.8 | 43.2 | 40.5 | 42.7 | 65.8 |
| DCAMA [29] | 41.9 | 45.1 | 44.4 | 41.7 | 43.3 | **69.5** | 45.9 | 50.5 | 50.7 | 46.0 | 48.3 | 71.7 |
| CyCTR [42] | 38.9 | 43.0 | 39.6 | 39.8 | 40.3 | 64.2 | 41.1 | 48.9 | 45.2 | 47.0 | 45.6 | 66.7 |
| DRCNet [5] | 44.8 | 53.2 | **49.5** | 45.7 | 48.3 | 67.7 | 50.2 | 57.7 | **52.0** | 50.1 | 52.5 | 68.5 |
| AAFormer [35] | 40.4 | 44.1 | 43.5 | 38.4 | 41.6 | 67.7 | 45.2 | 51.6 | 46.1 | 44.7 | 46.9 | 68.2 |
| VAT [8] | 39.1 | 42.7 | 42.9 | 40.5 | 41.3 | 68.8 | 46.5 | 47.3 | 48.0 | 49.7 | 47.9 | 72.4 |
| DPCN [17] | 42.0 | 47.0 | 43.2 | 39.7 | 43.0 | 63.2 | 46.0 | 54.9 | 50.8 | 47.4 | 49.8 | 67.4 |
| MIANet [40] | 42.7 | 52.9 | 47.8 | 47.4 | 47.7 | 67.1 | 45.8 | 58.2 | 51.3 | 51.9 | 51.6 | 68.3 |
| DiffSeg (ours) | **45.2** | **54.1** | 47.9 | **48.3** | 48.9 | 69.0 | 50.7 | 58.9 | 51.6 | 52.4 | 53.4 | **72.6** |

*COCO 2014* [15] is a challenging large-scale dataset containing 80 categories. The purpose of COCO is scene understanding, which is mainly acquired from complex daily scenes. The target objects in images are annotated in precise pixel-level masks.

According to the introduction of few-shot segmentation task, a cross-validation experiment is performed. On PASCAL-$5^i$, three splits are utilized for training, and the final split is used to evaluate models. On COCO 2014, according to [41], we select 40 classes for training, 20 classes for validation and 20 classes for test.

The mean Intersection-over-Union (mIoU) of all classes and the average of the foreground & background IoU (FB-IOU) are two main evaluation metrics of few-shot segmentation. For fair comparison with current methods, we use mIoU and FB-IOU to measure our model performance, but mIOU is regarded as the main metric due to its higher evaluation ability.

## 5.2 Implementation Details

To evaluate the performance of our method, we implement DiffSeg using the PyTorch library, and train it for 200 epochs on 4 Nvidia V100 GPUs. We set the learning rate to 0.01 and use the StepLR scheduler in PyTorch, thus reducing the learning rate to 0.9 times for every 20 epochs. Diffusion and Clip are pretrained models, which parameters are frozen in training phase. We only update parameters of PAM and SCM with a gradient descent algorithm. To reduce the effect of the random selection of support images, we run all tests 10 times and report the mean results.

## 5.3 Comparison with Other Methods

**PASCAL-$5^i$.** We compare DiffSeg with current methods on the PASCAL-$5^i$ dataset, where Table 1 shows results with mIOU and FB-IOU metrics. By comparing results, DiffSeg outperforms current methods and reaches new state-of-the-art performance in most cases. Under mIOU metric, DiffSeg is superior to current works in both 1-shot and 5-shot segmentation. We notice that DPCN [17] exceeds DiffSeg on split-3 of PASCAL-$5^i$, which is caused by nonuniform distribution of categories in splits. In fact, the average size of objects on split-3 is relatively large, where DPCN can better find the pixel-wise dense correlation between query and support images. Besides, considering that our approach is transformer-based, we compare DiffSeg with transformer-based methods (e.g., AAFormer [35] and VAT [8]) for fair comparison. The comparison results show that our method outperforms existing transformer-based methods. Essentially, our method benefits from rich prior knowledge rather than solely relying on transformer architectures, which is not present in other transformer-based methods.

Under FB-IOU metric, DiffSeg still achieves outstanding performance. Since mIOU is regarded as the main evaluation metric, we only report the average FB-IOU of 4 splits. More detailed results are reported in the Supplementary.

***MS COCO.*** The results of DiffSeg and other methods on the MS COCO dataset are shown in Table 2, from which we can see DiffSeg reaches the highest performance on MS COCO. Regarding that PASCAL-$5^i$ contains 20 categories and MS COCO contains

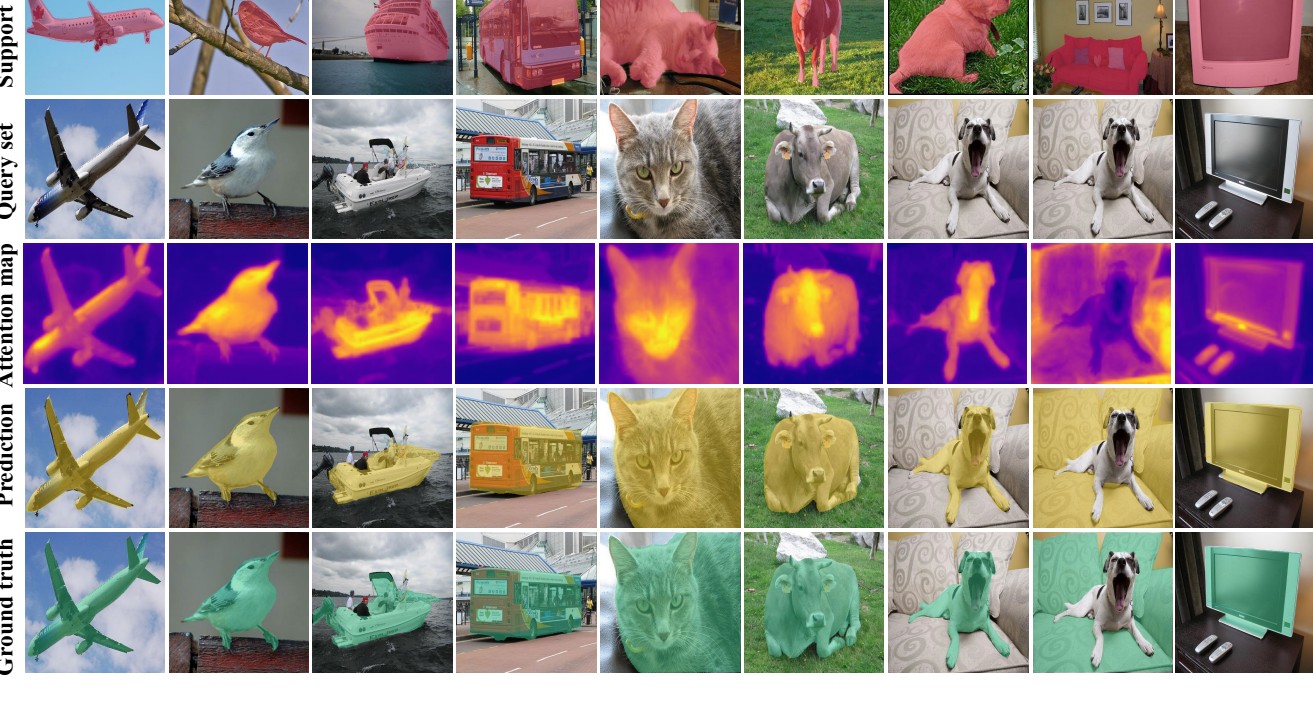

**Figure 5: The qualitative results of DiffSeg. The similarity between attention maps and predictions indicates that the knowledge of pretrained diffusion model provides a significant guidance for segmentation.**

**Table 3: Results of DiffSeg with different modules in 1-shot and 5-shot segmentation.**

| UNet | SCM | PAM | 1-shot | 5-shot |
|------|-----|-----|--------|--------|
| × | × | √ | 53.7 | 56.0 |
| √ | × | √ | 65.8 | 67.4 |
| √ | √ | × | 42.2 | 44.6 |
| √ | √ | √ | **69.3** | **72.1** |

**Table 4: Results of DiffSeg in 1-shot and 5-shot segmentation when different features are selected as prior knowledge representation.**

| Methods | SA | CA | 1-shot | 5-shot |
|---------|-----|-----|--------|--------|
| | √ | × | 59.4 | 62.7 |
| Attention | × | √ | 65.3 | 66.9 |
| | √ | √ | **69.3** | **72.1** |
| FRB | — | — | 50.1 | 52.8 |
| FST | — | — | 52.6 | 54.2 |

80 classes, the increase in classes brings more difficult problem and higher requirements to methods. As shown in Table 2, the evaluation score of mIOU drops significantly, but the score of FB-IOU is still at a high level. As most objects are small in images, even if models fail to predict target objects, the background IoU is still very high, thus resulting in a high FB-IOU.

**Qualitative results.** Some qualitative results of 1-shot segmentation are shown in Fig. 5, where each column represents the support set, query set, attention map of the pretrained diffusion model, prediction, and ground-truth, respectively.

Noted that given the same query image, DiffSeg is able to segment different targets when different objects are labeled in support images. For instance, in the 7th and 8th examples, given different support images with the category "dog" or "sofa", DiffSeg can segment different objects in the same query image. Essentially, the pretrained diffusion model can generate different attention score maps of a query image, which provides a strong guidance for segmenting different objects.

## 5.4 Ablation Study

To validate the effectiveness of each module, we perform ablation experiments on PASCAL-$5^i$, where we still use the cross-validation method and report the average mIOU of 4 splits.

**Network Design.** To prove effectiveness of the proposed modules, we compare the performance of DiffSeg when certain modules are removed. When UNet with SCM is removed, PAM only compares support latent images and query latent images, without considering attention maps. When PAM is removed, a convolutional layer is used to generate predicted latent image $L_{pre}$.

Table 3 shows the elimination of certain module would reduce the performance of DiffSeg, which means that each module plays a positive role in segmentation. When diffusion UNet is removed, segmentation performance drops significantly, which proves that prior knowledge in the pretrained diffusion model is powerful to

**Table 5: Results of DiffSeg in 1-shot and 5-shot segmentation when certain attention operations are removed or changed in PAM. "×" and "○" present removing attention and changing attention to convolutional operations, respectively.**

| Att_1 | Att_2 | Att_3 | 1-shot | 5-shot |
|-------|-------|-------|--------|--------|
| × | × | × | 44.3 | 46.5 |
| × | √ | √ | 58.6 | 60.4 |
| √ | × | √ | 52.1 | 53.7 |
| √ | √ | × | 64.2 | 66.8 |
| ○ | ○ | ○ | 59.2 | 61.1 |
| ○ | √ | √ | 62.4 | 63.7 |
| √ | ○ | √ | 59.9 | 61.5 |
| √ | √ | ○ | 65.4 | 67.2 |
| √ | √ | √ | **69.3** | **72.1** |

**Table 6: Results of DiffSeg in 1-shot and 5-shot segmentation when SCM is inserted into different levels of diffusion UNet.**

| 8×8 | 16×16 | 32×32 | Params sharing | 1-shot | 5-shot |
|-----|-------|-------|----------------|--------|--------|
| × | √ | √ | √ | 62.5 | 65.7 |
| √ | × | √ | √ | 60.3 | 62.1 |
| √ | √ | × | √ | 67.9 | 70.2 |
| √ | √ | √ | × | 69.2 | **72.1** |
| √ | √ | √ | √ | **69.3** | **72.1** |

interact semantic information between support-query images and provide a strong guidance for segmentation.

***Knowledge representation of diffusion models.*** To validate the significance of extracted prior knowledge, we compare results of DiffSeg which extracts feature maps at different locations as knowledge representation. First, we explore the performance impact of removing self-attention or cross-attention. Second, we study the impact of extracting features at other locations in diffusion UNet as prior knowledge, e.g., Features of Resnet Blcok (FRB) and Features of Spatial Transformer (FST).

The ablation results are as shown in Table 4, where SA and CA refer to self-attention and cross-attention, respectively. The results show that extracting only self-attention or cross-attention will result in a performance decrease. Self-attention maps effectively capture the similarity relationship between pixels of query images, and cross-attention maps contain the response information of query images to support images, which could contribute to extract semantic features and locate target areas. Noted that the result of DiffSeg with cross-attention is better than its with self-attention, which shows that information interaction between support images and query images is more important than semantic perception in query images. Besides, when using features of resnet block or spatial transformer as prior knowledge, the performance of DiffSeg decreases, which shows that attention could provide the fullest prior knowledge.

***Attention in PAM.*** To validate the effectiveness of 3 attention operations in PAM, we compare the performance of DiffSeg where certain attention operations are removed or changed to convolutional operations. The comparison results are shown in Table 5, where Att_1 and Att_2 present to cross-attention of query-support

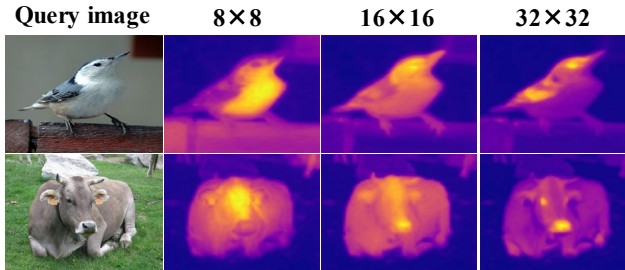

**Figure 6: Attention maps of different levels, where maps in 16×16 scale are the most complete.**

pairs and query-attention pairs, respectively. Att_3 presents a self-attention operation in PAM. From Table 5 we can see, removing certain attention operation could reduce the performance of DiffSeg, which signifies all attention operations in PAM play positive roles in segmentation. When Att_3 (self-attention) is removed, the model encounters difficulties in preserving a balanced segmentation dependency between the guidance from diffusion and support images. In cases these two sources of guidance are incongruent, accurate segmentation cannot be attained. It's worth noting that DiffSeg with Att_2 performs better than it with Att_1, indicating that spatial information from diffusion is more crucial than semantic information from support images for segmentation. Besides, the result of DiffSeg with attention operations is better than its with convolutional operations, which shows that attention mechanisms have more strong ability to capture interest areas for segmentation.

***Design of SCM.*** To validate advantages of SCM design, we insert SCM into different levels of diffusion UNet and compare the performance whether SCMs from different levels share parameters. The comparison results are shown in Table 5.

From the Table we can see, removing SCM from each level will lead to drop in performance, which shows SCM could work well in each level of diffusion UNet. Specifically, when SCM is deactivated in 16×16 scale level, performance of DiffSeg significantly decreases. Through visualizing attention maps of each level as shown in Fig. 6 , we find 16×16 attention maps are the most complete. Therefore, controls of 16×16 attention maps by SCM directly affect the performance of segmentation. Besides, training a separate SCM for each layer doesn't improve performance, while introducing additional parameters and increasing computational burden. It shows sharing parameters of SCM is a better strategy for lightweight computation.

## 6 CONCLUSION

In this paper, we present a DiffSeg for FSS, which decouples probabilistic denoising and segmentation processes, thereby mitigating the impact of probabilistic processes while benefiting from rich prior knowledge of the model. We propose a PAM for segmentation, where two cross-attention capture semantic information of support-query interaction and spatial information produced by the pretrained diffusion model. Moreover, considering the probabilistic generation of diffusion models, we present an SCM to align semantic boundaries between extracted attention maps and query images. Comprehensive experiments show that DiffSeg achieves new state-of-the-art performance.

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
