# OpenReview forum: "Few-shot Semantic Segmentation via Perceptual Attention and Spatial Control"
_acmmm.org/ACMMM/2024/Conference — MM2024 Poster_

### Official Review · Reviewer_M7Vb · 2024-05-14

**Rating:** 4
**Confidence:** 3

**Summary:**

This paper presents a new way to apply diffusion model in the task of Few-shot semantic segmentation (FSS). The spacial control module and the perceptual module are designed to encode the semantic information via the diffusion model and decode the semantic classification, respectively. Experimental results show that the proposed method can outperform previous SOTA methods on multiple metrics.

**Strengths:**

+ The presentation of this paper is good.

+ The way of how to apply diffusion models into FSS is novel, although there exist many works about diffusion-model-based semantic segmentation.

+ The visualization is clear and enough. The good and failure cases show in the main paper and the supplementary materials can provide enough information for the researchers in this community.

**Limitations:**

- One of our main concerns is what the role CLIP plays in the proposed framework? Intuitively, we think the semantic information from support images are important. According to the pipeline, the CLIP seems to help to encode the semantic information of support images. After checking the related works, we find that CLIP is rarely used. So, using CLIP may be a contribution? But the discussion about CLIP is missing. We want to see more discussions about the role of CLIP. If there is an ablation study about CLIP, this concern can be clearly solved.

- Although we find that previous SOTAs rarely provide the information about computational cost, we still want to know the inference cost (time and FLOPS) of the proposed method. Because the proposed framework includes both the CLIP and UNet of diffusion model, we think the computational cost may be large.

- Why to choose edge maps for SCM? Can other conditions help? Discussions about this choice may be expected.

- Citation information should be updated: Citation 12 has been accepted by AAAI 24.

**Suitability:**

3

---

### Official Review · Reviewer_2YnP · 2024-05-24

**Rating:** 5
**Confidence:** 3

**Summary:**

This paper presents a model called DiffSeg, which leverages prior knowledge from diffusion models to assist in few-shot semantic segmentation tasks. To address the issues of semantic and spatial information inconsistency and the uncertainty in the generation process, the paper proposes the Perceptual Attention Module (PAM) and the Spatial Control Module (SCM) respectively. Experiments have demonstrated that DiffSeg achieves competitive results in few-shot semantic segmentation tasks.

**Strengths:**

1. The paper proposes utilizing edge information to enhance the diffusion model's ability to preserve the original shape, thereby increasing the certainty of the generated segmentation masks.
2. The number of modules that need to be trained is relatively small. Although the model uses CLIP and diffusion models, only PAM and SCM need to be trained to adapt these features, alleviating the difficulty of fine-tuning the pre-trained models.
3. The paper is well-written, and the experiments are thorough, demonstrating the validity of the model.

**Limitations:**

1. Compared to the original features, what does the Spatial Control Module learn? The model keeps the parameters of the diffusion model completely frozen, yet the features change after passing through the Spatial Control Module, leading to domain shift. Explaining the changes that the features undergo can better elucidate the role of edge information and SCM.
2. Compare the model size and efficiency with other related works. In semantic segmentation tasks, it is crucial to balance performance and efficiency. DiffSeg employs both diffusion models and CLIP models, making it necessary to compare metrics such as model size and inference time.

**Suitability:**

2

---

### Official Review · Reviewer_2WhW · 2024-05-29

**Rating:** 4
**Confidence:** 4

**Summary:**

This article applies diffusion models to FSS tasks, in which it is difficult to maintain spatial relationships between inputs and outputs due to probabilistic noising and denoising processes. To address this issue, a diffusion-based Segmentation network is proposed.

**Strengths:**

The novelty is to decouple probabilistic denoising and segmentation processes to maintain spatial relationships between inputs and outputs. And the experiments on PASCAL-5𝑖 and COCO datasets show that the method proposed achieves the state-of-the-art performance.

**Limitations:**

1) The effect has not been significantly improved.
2) The problem is that it is difficult to maintain spatial relationships between inputs and outputs due to probabilistic noising and denoising processes. The experimental results do not indicate whether the problem has been solved.

**Suitability:**

2

---

### Official Review · Reviewer_4UHB · 2024-06-02

**Rating:** 4
**Confidence:** 3

**Summary:**

The paper proposes a framework to leverage pre-trained diffusion models in the segmentation problem.
DiffSeg is proposed for Few-shot Segmentation (FSS) problem to tackle the issues of probabilistic processes within current diffusion-based segmentation models and further utilize the prior knowledge from existing diffusion models for segmentation

+ The authors propose a Perceptual Attention Module (PAM) for segmentation, which comprises a cross-attention module for capturing the semantic and spatial information obtained from the supporting query and the diffusion model, respectively. The output is post-processed by a self-attention mechanism to balance out the weight of each cross-attention module previously.
+ The authors propose a Spatial Control Module (SCM) for aligning the semantic boundaries between the input query images and the extracted attention maps, which bridges the gap between the spatial structures of the input and output features obtained from the diffusion model.

The model's performance is reported to be the current SOTA for the FSS task, as it achieves 69.3% for 1-shot segmentation and 72.1% for 5-shot segmentation.

**Strengths:**

The proposed idea of segmentation as a conditioned generation with diffusion models is creative.

The architecture mostly relies on pre-trained models and only introduces a few additional modules, which require fewer computational resources to train and preserve the utility of the pre-trained models.

The modules have clear functionalities. The SCM module is well-motivated and easy to understand.

The methods of the authors have been experimented and compared under various settings, with the proof of efficiency and accuracy of the
model by stress-testing at the end of the Experiments section.

**Limitations:**

The presentation is somewhat confusing. In Figure 2, trying to include the training losses makes the pipeline cluttered.

Details regarding the VAE seem not mentioned anywhere. Where are their parameters coming from?

The authors mentioned that their PAM module is different from previous works. However, they do not justify the differences in detail enough. Given that the authors' idea is interesting, its implementation warrants a more in-depth discussion and possibly more experiments to justify the design choices, rather than traditional ablation studies.

To formally support the claim of the paper's main purposes, the authors should describe the problems of uncertainties in a more formal fashion and prove or support that the model can help segregate the process from the segmentation phase.

**Suitability:**

2

---

### Meta-Review · Area_Chair_dWx8 · 2024-06-26

**Recommendation:** Accept (Poster)
**Confidence:** 4

**Metareview:**

The work is of reasonable quality --there are some concerns about the limited novelty, yet is has merits.